# Debate on the Role of Eccentric Contraction of the Diaphragm: Is It Always Harmful?

**DOI:** 10.3390/healthcare13050565

**Published:** 2025-03-05

**Authors:** Adrián Gallardo, Mauro Castro-Sayat, Melina Alcaraz, Nicolás Colaianni-Alfonso, Luigi Vetrugno

**Affiliations:** 1Department of Kinesiology and Respiratory Care, Sanatorio Clínica Modelo de Morón, Morón C1708, Argentina; 2Departamento de Ciencias de la Salud, Kinesiología y Fisiatría, Universidad Nacional de la Matanza, San Justo B1754, Argentina; 3Non-Invasive Respiratory Care Unit, Juan A. Fernández Hospital, Buenos Aires C1425, Argentina; maurocastrosayat@gmail.com (M.C.-S.); nicolkf@gmail.com (N.C.-A.); 4Department of Medical, Oral and Biotechnological Sciences, University of G. d’ Annunzio, 66100 Chieti, Italy; luigi.vetrugno@unich.it

**Keywords:** diaphragm, ventilator weaning, reverse triggering, eccentric contraction, mechanical ventilation

## Abstract

The diaphragm is the primary muscle involved in the ventilatory pump, making it a vital component in mechanical ventilation. Various factors in patients who require mechanical ventilation can lead to the deterioration of the diaphragm, which is associated with increased mortality. This deterioration can arise from either excessive or insufficient support due to improper adjustment of ventilation programming variables. It is essential for healthcare professionals to make appropriate adjustments to these variables to prevent myotrauma, which negatively impacts muscle structure and function. One recognized cause of muscle injury is eccentric work of the diaphragm, which occurs when muscle contractions continue after the expiratory valve has opened. Current evidence suggests that these eccentric contractions during mechanical ventilation can be harmful. This brief review highlights and analyzes the existing evidence and offers our clinical perspective on the importance of properly adjusting ventilation programming variables, as well as the potential negative effects of eccentric diaphragm contractions in routine clinical practice.

## 1. Introduction

The diaphragm is the primary muscle involved in breathing and plays a crucial role in ensuring adequate alveolar ventilation. Various factors can impair diaphragm function in patients who require mechanical ventilation (MV). These impairments often arise from excessive unloading or insufficient assistance, specifically through over-assistance or under-assistance, which can result from inappropriate ventilator settings. Both scenarios can lead to harmful changes in the diaphragm’s structure and function [1].

Although MV is intended to support or replace the respiratory muscles, the main mechanisms contributing to diaphragm myotrauma are typically linked to inadequate ventilatory adjustments. It has been suggested that in the early stages of critical illness, diaphragm weakness—often caused by ventilatory over-assistance—can mimic organ failure and is associated with increased mortality [2]. Factors such as inadequate levels of positive end-expiratory pressure (PEEP), excessive or insufficient assist pressure or volume, and the presence of asynchronies, including eccentric diaphragmatic contractions, must be considered. These issues can damage muscle fibers, alter contractile function, and lead to weakness or atrophy, which are also associated with longer stays in the intensive care unit (ICU), a higher risk of complications, and increased mortality [1,2,3].

Patient–ventilator interactions are crucial in invasive mechanical ventilation, and a lack of synchrony has been linked to poor outcomes. Diaphragmatic myotrauma refers to adverse interactions between the patient and the ventilator, leading to diaphragm atrophy and injury, ultimately causing weakness and dysfunction. Clinical studies indicate that after just 24 h of mechanical ventilation, 64% of patients exhibit diaphragmatic weakness, with average strength reductions of 32% ± 6% by day six of MV, alongside decreased cross-sectional area and increased expression of protein degradation enzymes. Up to 80% of patients may experience this weakness by the time they are weaned from ventilation during their ICU stay [1,4,5]. Such dysfunction can greatly affect clinical outcomes, as diaphragmatic weakness increases the risk of sustained respiratory failure, significantly delaying the weaning process [3].

These considerations highlight the need for ventilatory strategies that protect both the lungs and the diaphragm. Understanding the potential benefits and drawbacks of eccentric diaphragm work is essential, as it could lead to new insights into diaphragm function during mechanical ventilation and ultimately improve patient outcomes in critical care. To present our point of view, we performed a literature search on Medline, PubMed, SciELO, and Google Scholar using the keywords presented above as search parameters. We then included information considering the availability of full texts and those in English. Finally, three authors selected the texts according to their relevance to develop the point of view in a comprehensive and concise manner.

## 2. Mechanisms of Diaphragmatic Injury

### 2.1. Level of Assistance

One potential risk associated with excessive ventilatory support is myotrauma due to over-assistance. When ventilatory support is too high, it can diminish the patient’s respiratory drive, leading to impaired contraction of the diaphragm from disuse atrophy [5,6,7]. In this scenario, the diaphragm has fewer contractile units available to perform the same mechanical work, making it more susceptible to reduced function and early ventilatory failure when support is eventually decreased [3,8]. Conversely, insufficient assistance is responsible for elevated respiratory drive and excessive respiratory muscle activity, leading to sarcolemma rupture and diaphragmatic dysfunction [8,9]. Additionally, the negative pleural pressure generated by diaphragmatic contraction is not uniformly transmitted to the lungs, which can result in lung injury. According to solid dynamics principles, this uneven distribution causes a pendelluft phenomenon [3,10,11,12]. This phenomenon exposes the dorsal lung regions to detrimental tidal volume distribution as the ventral regions empty during inspiration. Consequently, it is essential to appropriately program mechanical ventilation: excessive inspiratory effort may lead to overdistension of the dependent lung regions, potentially resulting in injury. This underscores the significance of meticulously managing inspiratory effort to prevent such damage.

### 2.2. Positive End-Expiratory Pressure and Asynchronies

Application of positive end-expiratory pressure (PEEP) is a standard practice in mechanical ventilation (MV) designed to enhance gas exchange and improve respiratory mechanics by increasing the end-expiratory lung volume (EELV). However, the application of excessive PEEP can contribute to myotrauma, as the resultant increase in end-expiratory lung volume (EELV) may displace the diaphragm downward. This downward movement flattens the diaphragm’s dome, reduces the apposition zones, and positions the diaphragm in a less optimal configuration for generating force. While these potential downsides exist, the benefits of PEEP should not be overlooked [9,13,14].

The prolonged displacement can favor longitudinal atrophy, resulting in the loss of sarcomeres in series. This leads to alterations in the diaphragm’s length–tension relationship, affecting its contractile function and neuromechanical efficiency [15]. Conversely, reducing PEEP (for instance, during T-tube trials) can significantly decrease EELV. This may cause excessive stretching of the diaphragm fibers, which can move cephalically and create a new mechanical environment. Such stretching can impair contractile force generation due to the mechanical disadvantages of excessive elongation and longitudinal atrophy [9,13,14].

Another harmful mechanism that can arise during MV is eccentric contraction. This occurs when the ventilator begins the expiratory phase (by opening the expiratory valve) before the neural inspiratory phase has completed. This post-inspiratory load can happen during various types of patient–ventilator asynchronies, such as reverse triggering or early cycling, as well as in circumstances involving expiratory braking. These conditions prevent alveolar collapse and reduce the risk of atelectasis by decreasing peak expiratory flow [16].

Patient–ventilator asynchronies arise from a mismatch between the patient’s respiratory demands (in terms of magnitude, speed, and timing of contraction) and the mechanical assistance provided (in terms of magnitude, speed, and timing). Such asynchronies can lead to vigorous diaphragmatic contractions during the expiratory phase [3], potentially resulting in lung injury [17,18]. Types of asynchronies that may produce these harmful contractions include reverse triggering, ineffective effort, and premature cycling (Figure 1). Patients experiencing acute respiratory failure often exhibit elevated respiratory drive, making them more susceptible to vigorous contractions. A recent study suggested that patients whose efforts resemble those of healthy individuals at rest tend to have better outcomes than those with either stronger or weaker efforts [19].

In cases of reverse triggering, for example, diaphragm contraction is caused by passive mechanical insufflation, which usually occurs in sedated patients or those receiving high levels of ventilatory assistance [20]. Muscular contraction happens after a mandatory ventilator cycle, leading to eccentric loading, either with or without breath stacking. Both scenarios are potentially injurious [20,21,22,23]. These phenomena are recognized in MV practices and are linked to progressive deterioration of diaphragmatic function [15].

However, it is important to adopt a more holistic perspective, acknowledging that the effects on the diaphragm can vary based on context (e.g., the patient’s susceptibility, underlying condition, effort magnitude, and effort duration). Under some circumstances, eccentric contraction combined with adequate respiratory effort might not have harmful effects. In fact, it can enhance diaphragmatic strength by preventing disuse and atrophy [21,23,24]. Conversely, if these contractions occur with excessive effort or neuromechanical delays, they may damage diaphragm structure and function, causing pulmonary and muscular injuries [2,22,23] (Figure 2).

In an experimental study using an animal model of acute respiratory distress syndrome (ARDS), reverse triggering impacted diaphragmatic function differently depending on the level of respiratory effort. The combination of reverse triggering and elevated inspiratory effort was associated with worsening diaphragmatic function and structure after just three hours, resulting in greater muscular injury [23]. These findings underscore the importance and challenge of monitoring contraction magnitudes, particularly during incidents of reverse triggering; however, this has been difficult to achieve without advanced monitoring devices or algorithms [24].

In two studies involving patients with ARDS, reverse triggering was found to occur in over 50% of subjects within the first 24 h of mechanical ventilation. Interestingly, one of the studies indicated that patients who experienced a reverse trigger showed better oxygenation and were more likely to be converted to assisted ventilation and extubated within 24 h [21]. Furthermore, in the other study, patients had lower in-hospital mortality rates, suggesting the possibility of favorable or at least non-deleterious outcomes [25].

These findings are not entirely surprising, as it is understood that diaphragmatic contraction may function as a brake, delaying or reducing expiratory collapse while maintaining pulmonary aeration more effectively than mechanical ventilation combined with muscle paralysis and no muscle activity [16].

## 3. Post-Inspiratory Contraction and Mechanical Implications

The gold standard for monitoring respiratory effort is the use of esophageal pressure (Pes) or trans-diaphragmatic pressure (Pdi), owing to their precision and reliability. While Pes provides a complete assessment of all inspiratory muscles, the calculation of Pdi requires both gastric and esophageal pressure measurements.

Inspiratory muscular pressure (Pmus) represents the overall activity of the inspiratory muscles, accounting for the elastic recoil of the thoracic cage. It can be expressed by the equation: [26]

[Pmus = Pcw − Pes]

In this equation, Pcw denotes the chest wall pressure. This equation can be further divided into its mechanical components (elastic and resistive) as follows:

[Pmus = PEEPi + (Elrs × Vt) + (Raw × Flow)]

In this expression, PEEPi refers to intrinsic PEEP, Elrs represents elastance, Vt is the tidal volume, and Raw indicates airway resistance. During mechanical ventilation, the ventilator assistance (Pvent) must be incorporated:

[Pvent + Pmus = PEEPi + (Elrs × Vt) + (Raw × Flow)]

It is essential to consider both the contraction time and magnitude, as these factors directly relate to oxygen consumption (metabolic cost). One method of evaluating this is through the temporal integral of Pmus, known as the pressure–time product (PTP), which can be measured as either esophageal (PTPmus) or diaphragmatic (PTPdi):

[PTP = Pressure × Time = cmH_2_O × sec]

However, during eccentric contractions, calculating PTPmus (post-inspiratory) using traditional methods is challenging due to the absence of a temporal flow reference. This limitation may lead to an underestimation of PTP per minute and the work of breathing (WoB). To evaluate the load imposed on diaphragm fibers that elongate at the start of the expiratory phase, the Campbell diagram and partitioning of the Pdi–volume loop can be utilized (as shown in Figure 3). This work can be expressed in terms of negative work (J/L) [27] (Figure 3). However, employing this method requires specialized software and algorithms for real-time monitoring [25].

In cases where reference methods (such as Pes, Pdi, or electrical activity of the diaphragm (EAdi)) are unavailable, ventilator signals (flow, pressure, volume) can effectively indicate the presence of Pmus and provide insight into its behavior in relation to Pvent [28].

These mechanical considerations are particularly pertinent in clinical conditions such as COPD and ARDS. In patients with COPD, the diaphragm is frequently found in a more caudal position, resulting in a shortened configuration that diminishes its mechanical advantage for generating force. In cases of ARDS, although the diaphragm may be aligned in a “physiological” position, the force generated during contraction may not effectively promote ventilation. This is due to the potential obstruction of airflow in certain lung regions, which is influenced by overlapping pressures (including both elastic and non-elastic resistance). In other words, when force generation is compromised by the underlying pathophysiology of these diseases, vigorous eccentric contraction may further exacerbate inefficiencies, leading to an increased metabolic cost and a range of detrimental consequences.

## 4. Controversy

It is widely recognized that during total ventilatory support, the accurate programming of mechanical ventilation is vital. This includes adjusting tidal volume and minute ventilation in accordance with predicted body weight, with the objective of achieving a plateau pressure of 28 cmH_2_O or lower, as well as individualizing PEEP and maintaining a driving pressure below 15 cmH_2_O. However, the discourse concerning the potential benefits and drawbacks of eccentric contractions remains a topic of ongoing debate. When respiratory effort is high, it can lead to eccentric myotrauma, muscle fiber injury, and altered muscle function. This highlights the importance of monitoring and adjusting respiratory drive and effort as necessary. However, it is also important to note that adequate respiratory effort during eccentric contractions may in some instances help preserve diaphragmatic function under specific conditions [21,22,23,25]. In the context of limb muscles, periodized training that includes eccentric contractions has shown positive effects. However, when it comes to diaphragmatic training, the optimal application of intensity, volume, and frequency of training remains unclear [22]. A novel approach involving phrenic nerve stimulation has been suggested to prevent muscle dysfunction due to disuse [29]. This neurostimulation may help maintain muscle strength and counteract the compressive pressure exerted by abdominal contents. Additionally, it promotes a more physiologically normal distribution of tidal volume, which helps keep the dorsal alveoli open by delivering more tidal volume to them [30,31]. Despite these promising results, further research on treatment outcomes is necessary, and audience engagement is crucial for the future of strategies aimed at protecting the diaphragm [31].

Conversely, in patients in the early phase of severe ARDS, there exists a discrepancy between lung-protective ventilation and diaphragmatic protection [10]. In this context, neuromuscular blockade during the first 48 h can improve outcomes, such as mortality, by minimizing injuries associated with patient–ventilator interactions due to diaphragmatic inactivity [5,9]. However, the ROSE study [32], which was halted before fully recruiting the planned number of participants, reported that the use of cisatracurium within the first 48 h of ARDS had no favorable impact on hard outcomes (e.g., mortality) at 90 days, with similar results observed in both groups evaluated. This raises questions about prioritizing patient–ventilator interaction, even in the early stages of ARDS.

In summary, the effect of eccentric contractions has not yet been fully elucidated. It is essential to adopt a therapeutic approach aimed at maintaining an adequate level of inspiratory effort and drive in patients on mechanical ventilation [3,8,19] (Figure 4).

A reliable estimation of respiratory effort should be sought and appropriate adjustments to ventilator settings should be made to maintain adequate patient–ventilator interaction [33]. Tools such as esophageal pressure monitoring, diaphragmatic electrical activity assessment, ultrasound, and measurement of respiratory drive through P0.1 and ∆Pocc are useful for evaluating the magnitude of drive and inspiratory effort to optimize patient–ventilator interaction and avoid diaphragmatic dysfunction. The latter has been shown to be frequent in patients requiring mechanical ventilation and is associated with prolonged weaning, higher complication rates, and higher mortality. Esophageal pressure measurement provides a direct and accurate assessment of respiratory effort, while monitoring diaphragmatic electrical activity offers an indirect measure of respiratory drive [27,32,34]. Ultrasound can visualize the diaphragm and assess its structure and function [35,36], as well as help assess the interaction between the patient and ventilator [37] (Figure 5 and Table 1).

## 5. Conclusions

Eccentric diaphragmatic contraction is a clinical reality frequently observed in association with asynchronies. Although current evidence suggests avoiding this type of muscular work, other studies have shown a beneficial effect when the intensity of the contraction is close to normal values. A priori, it is logical to think that this activity might not always be harmful, but on the contrary, it could preserve the trophism and activity of the diaphragm, being positive in terms of weaning from mechanical ventilation. Consequently, robust studies are required that can provide clinical data to determine the true role of this type of diaphragmatic work.

## Figures and Tables

**Figure 1 healthcare-13-00565-f001:**
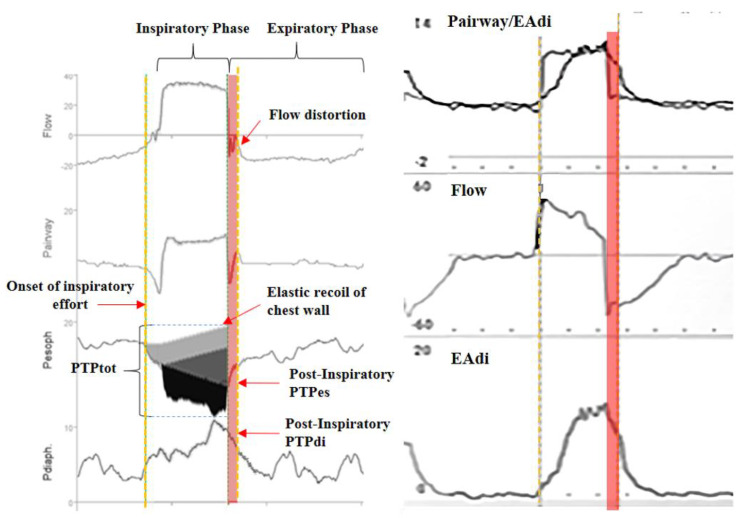
Post-inspiratory loading in premature cycling (Pes and EAdi). PTPtot: sum of the product of the time pressure performed by the patient and by the ventilator. The pink bands show the period during which the diaphragm works eccentrically. PTPes: esophageal time pressure product; PTPdi: transdiaphragmatic time pressure product; EAdi: electrical activity of diaphragm; Paw: airway pressure.

**Figure 2 healthcare-13-00565-f002:**
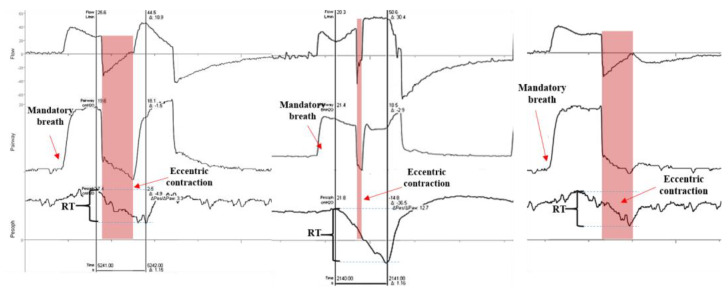
Different magnitude and speed of contraction during reverse triggering with and without breath stacking. The pink bands indicate the period during which the diaphragm works eccentrically. RT: reverse triggering.

**Figure 3 healthcare-13-00565-f003:**
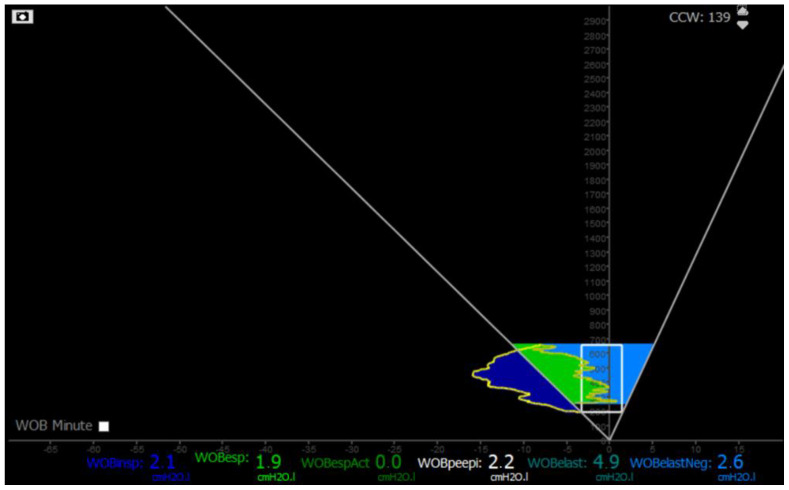
Campbell diagram showing elastic negative work of breathing (light blue) due to diaphragm eccentric contraction. Blue zone: inspiratory work of breathing. Green zone: expiratory work of breathing. White square: work of breathing associated to intrinsic PEEP. WoB: work of breathing.

**Figure 4 healthcare-13-00565-f004:**
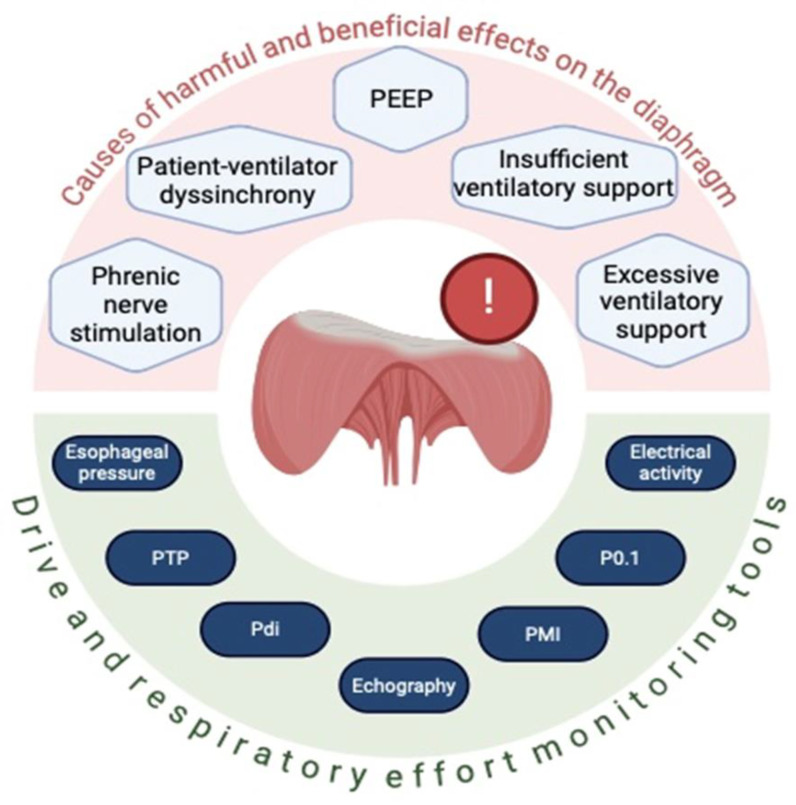
Causes of harmful and beneficial effects of diaphragm eccentric contraction and respiratory drive monitoring options. PTP: pressure–time product. PMI: pressure muscle index. P0.1: occlusion pressure at the first 100 milliseconds of the respiratory cycle. ∆Pocc: occlusion pressure during tidal ventilation.

**Figure 5 healthcare-13-00565-f005:**
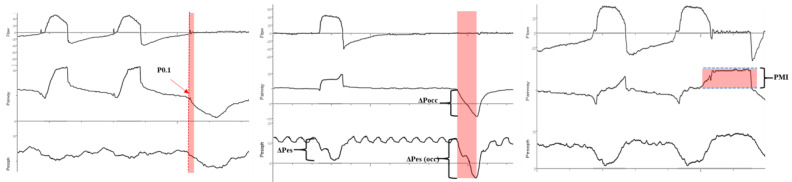
Non-invasive bedside methods for effort monitoring. The pink bands indicate the occlusion maneuver period. ΔPes: esophageal pressure in a tidal breath. P0.1: occlusion pressure in the first 100 milliseconds; ΔPocc: occlusion pressure in a tidal breath; PMI: pressure muscle index.

**Table 1 healthcare-13-00565-t001:** Methods for bedside effort monitoring.

Measurements	Utility	Limitation	Reference Values
P0.1	- Occlusion pressure measurement in the first 100 ms. - Allows monitoring of respiratory drive and detecting the presence of low or high respiratory effort. - Simple maneuver, available automatically on most ventilators.	- Increased respiratory drive does not always result in increased respiratory effort (i.e., in the presence of severe respiratory muscle weakness or short inspiratory time).	- 1–4 cmH_2_O in critical Ill patients
ΔPocc	- Assesses excessive respiratory effort and pulmonary stress. - Simple maneuver, requires only the performance of an expiratory pause. Available in most ventilators.	- -Assumes that the effort of the breaths prior to occlusion are equal.	- ΔPocc 8–20 cmH_2_O (estimated Pmus 5–10 cmH_2_O)
PMI	- Represents the interaction between inspiratory effort and lung compliance in tidal volume delivered during partial support.	- Requires relaxation of the expiratory muscles during occlusion at the end of inspiration.	- PMI > to 2 < to 6 cmH_2_O
Pes	- Assesses the magnitude of the contribution of all inspiratory muscles. - Allows temporal integration of the Pmus (PTPmus) and partitioning of it into elastic and resistive components. - Allows one to analyze and partition the work using esophageal pressure and volume (Campbell diagram). Negative work (J/L) can give information about eccentric contractions.	- Minimally invasive, requires dedicated software. - Requires expertise for placement and validation. - PTPes cannot be evaluated (in vivo) with the usual tools in the presence of eccentric contractions. Risk of underestimation of work of breathing.	- 5–10 cmH_2_O in healthy subjects breathing calmly. - Avoid Pes < 2–3 cmH_2_O. - PTP/min 50–150 cmH_2_O x sec/min in healthy subjects breathing calmly. - WoB 2.4–7.5 J/min or 0.2–0.9 J/L.
Pdi	- Allows one to estimate in isolation (by subtraction) the contribution of the diaphragm. - Allows temporary integration of Pdi (PTPdi). - Useful in the presence of expiratory effort.	- Minimally invasive, requires dedicated software. - Requires expertise (idem Pes). - Requires gastric balloon placement (in addition to esophageal balloon).	- 5–10 cmH_2_O in healthy subjects breathing quietly.
EAdi	- Allows dynamic monitoring of the neural drive. - Allows evaluation of the neuromuscular efficiency index (NME), normalizing EAdi to Pes, Pdi, or ΔPocc.	- Minimally invasive. - Requires a catheter and use of a specific ventilator. - Requires expertise for placement and validation.	- EAdi: 5–20 µv in critically ill patients (expert opinion) - NME: 0.5–2 cmH_2_O/µv in critically ill patients (expert opinion)
Diaphragmatic ultrasound	- Allows noninvasive assessment of diaphragmatic contractility (thickening fraction). - Non-invasive, simple, bedside measurement. - Provides an index of diaphragmatic contraction (tidal TFdi).	- Equipment and training required	- TFdi 15–30%

P0.1: occlusion pressure in the first 100 milliseconds; ΔPocc: occlusion pressure in a tidal breath; PMI: pressure muscle index; Pes: esophageal pressure; Pdi: transdiaphragmatic pressure; EAdi: electrical activity of diaphragm.

## Data Availability

Data sharing is not applicable. No new data were created or analyzed in this study.

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
