# Peer review of "Debate on the Role of Eccentric Contraction of the Diaphragm: Is It Always Harmful?"

_healthcare, 2025, doi:10.3390/healthcare13050565_

Round 1
Reviewer 1 Report
Comments and Suggestions for Authors
The manuscript addresses a critical aspect of patient care in mechanical ventilation, offering valuable perspectives for clinical practice. Below, I provide comments and suggestions for refinement:
1. more detailed discussion of the underlying mechanisms and evidence from recent studies would strengthen the argument;
2. provide brief framework to help clinicians adjust ventilation programming variables to mitigate the risks of diaphragm myotrauma;
3. include a discussion of the potential long-term consequences of diaphragm myotrauma (e.g., delayed weaning, increased ICU stay) and its impact on patient outcomes would enhance the manuscript's significance;
4. the language is clear and professional, but minor refinements could improve readability.
Reviewer 2 Report
Comments and Suggestions for Authors
It is very interesting manuscript but I have some remarks.
1. It is unclear which keywords and which databases were used to search the sources.
2. Please, format sources according to a single template (2,9,24,26, 27, 30-32, 34,36,37). 3. 5 sources out of 37 are over 10 years old. 4. The order of citations is out of order (20, 22-23, after that 21). 5. There is no reference 15 to the source in the text. 6. Abbreviations PEEP and ARDS entered twice. 7. Please, add explanation of abbreviations for all pictures and formulas.8. Table 1 contains a very large amount of information. Could you, please, reduce it or move it to section Suppl. materials?
Reviewer 3 Report
Comments and Suggestions for Authors
Congratulations for your manuscript trying to present your point of view regarding the eccentric contraction of diaphragm. However, I would prefer the presentation of your experience regarding your patients and any outcomes. Furthermore, I would also prefer a review even a narrative one due to the fact that several manuscripts have already been published regarding this topic. What is more, you present the existing literature and you also present the controversy which has not been resolved yet without mentioning your opinion after all. What I am trying to say is that the aim of a manuscript should be covering of a knowledge gap or presenting some original research findings. These would be more interesting for the audience. I am not sure that the type of the article you have chosen is an appropriate one. On the other hand, the manuscript is comprehensible and presents the existing literature. I suggest you rewrite the manuscript with the structure of a review, as you have already included some interesting studies.
Reviewer 4 Report
Comments and Suggestions for Authors
It was my pleasure to review your technical paper. Please find my comments below.
1. Introduction is well drafted and underscores the need for your review.
2. I would suggest explaining in more detail about diaphragm contractions in different disease states e.g COPD, ILD, ARDS to make your review more clinically relevant.
3. Please reference clinical studies where the mechanical implications have been studied or calculated.
4. Please mention how we can use the information clinically for better patient outcomes.
Thank you.
Round 2
Reviewer 3 Report
Comments and Suggestions for Authors
Congratulations for your work again. However, I have to insist on my initial review. Although the manuscript is comprehensible, I cannot overlook the fact that you did not cover any knowledge gap. On the other hand, the results of your protocol would be more interesting. I will the final decision to the Editor of course, but I have to admit that I am quite concerned about your manuscript.
